# Translation, Cross-Cultural Adaptation and Psychometric Validation of the Greek Version of the Cardiac Rehabilitation Barriers Scale (CRBS-GR): What Are the Barriers in South-East Europe?

**DOI:** 10.3390/ijerph20054064

**Published:** 2023-02-24

**Authors:** Varsamo Antoniou, Konstantinos Pasias, Nektarios Loukidis, Kalliopi K. Exarchou-Kouveli, Demosthenes B. Panagiotakos, Sherry L. Grace, Garyfallia Pepera

**Affiliations:** 1Clinical Exercise Physiology and Rehabilitation Laboratory, Department of Physiotherapy, School of Health Sciences, University of Thessaly, GR-35100 Lamia, Greece; 2Department of Nutrition and Dietetics, School of Health Sciences and Education, Harokopio University, GR-17671 Athens, Greece; 3School of Kinesiology and Health Science, Faculty of Health, York University, Τoronto, ON M3J 1P3, Canada; 4KITE Research Institute and Peter Munk Cardiac Centre, University Health Network, University of Toronto, Toronto, ON M5G 2A2, Canada

**Keywords:** Cardiac Rehabilitation Barriers Scale, cardiac rehabilitation, reliability, cross-cultural adaptation, psychometric validation, coronary artery disease, health care use

## Abstract

Cardiac Rehabilitation (CR) is a secondary prevention intervention proven to improve quality of life, yet with low participation. The Cardiac Rehabilitation Barriers Scale (CRBS) was developed to assess multi-level barriers to participation. This study aimed at the translation, and cross-cultural adaptation of the CRBS into the Greek language (CRBS-GR), followed by psychometric validation. Some 110 post-angioplasty patients with coronary artery disease (88.2% men, age 65.3 ± 10.2 years) answered the CRBS-GR. Factor analysis was performed to obtain the CRBS-GR subscales/factors. The internal consistency and 3-week test–retest reliability was evaluated using Cronbach’s alpha (α) and intraclass correlation coefficient (ICC), respectively. Construct validity was tested via convergent and divergent validity. Concurrent validity was assessed with the Hospital Anxiety and Depression Scale (HADS). Translation and adaptation resulted in 21 items similar to the original version. Face validity and acceptability were supported. Construct validity assessment revealed four subscales/factors, with acceptable overall reliability (α = 0.70) and subscale internal consistency for all but one factor (α range = 0.56–0.74). The 3-week test-retest reliability was 0.96. Concurrent validity assessment demonstrated a small to moderate correlation of the CRBS-GR with the HADS. The greatest barriers were the distance from the rehabilitation center, the costs, the lack of information about CR, and already exercising at home. The CRBS-GR is a reliable and valid tool for identifying CR barriers among Greek-speaking patients.

## 1. Introduction

Cardiovascular diseases (CVDs) remain one of the leading causes of morbidity and mortality worldwide [1,2], accounting for approximately 43% of all recorded deaths in Europe [3]. Data from the Hellenic Statistical Authority (HELSTAT) and the Global Burden of Disease (GBD) study show the high prevalence of CVDs (37% of all deaths) [4] and their increasing rates during the previous decade [5] amongst the Greek population. Additionally, CVDs are responsible for 27 million disability-adjusted life years (DALYs) in the European Union (EU); thus imposing an approximately €210 billion burden on the EU economy [6].

Secondary prevention interventions can mitigate the aforementioned burden by reducing recurrent cardiovascular events [7]. Recent guidelines on CVD prevention recommend cardiac rehabilitation (CR), which is a secondary intervention delivering exercise training, dietary counseling, smoking cessation, cardiac risk factor modification and psychosocial support by a multidisciplinary team [8,9]. Despite the well-established effectiveness of CR in enhancing CVD patients’ overall well-being and quantity of life [10,11,12,13], CR participation rates remain discouragingly low.

There is also inequity in CR use, with female patients, those of older age and lower socioeconomic status, i.e., education level and hyperlipidemia even less likely to use CR. Barriers to CR participation are multi-level, including factors such as lack of clinician referral, travel distance, work and family responsibilities as well as a lack of motivation [14,15,16]. Moreover, additional barriers were precipitated by the COVID-19 pandemic, such as the suspension of center-based CR programs and the implementation of infection control measures [17,18,19].

Even in high-income countries CR utilization rates are discouragingly low [20]. In Greece specifically, it is estimated that less than 1% of CVD patients are referred and hence have access to outpatient CR programs [1,21]. With only four CR programs available in Greece [21], it is clear that there is an urgent need for CR development. Yet, to our knowledge, there is no systematic study of CR barriers in the country. Validated scales such as the Cardiac Rehabilitation Barriers Scale (CRBS)—which assesses perceptions of barriers to CR enrollment and participation from the patient to health system levels–can support identification and hopefully ultimately lead to a mitigation of barriers [22]. The main purpose of this study was to translate and cross-culturally adapt the CRBS to the Greek language, and then psychometrically validate it. The secondary aim was to identify the main CR barriers in the Greek cardiac population.

## 2. Methods

### 2.1. Design

A multi-step process was followed to translate, cross-culturally adapt and psychometrically validate the Greek version of the CRBS (CRBS-GR). The translation and cross-cultural adaptation of the scale were carried out in accordance with best practice recommendations [23,24]. The psychometric validation was a cross-sectional study, but with a 3-week test-retest interval in a subset of participants. Data for this aspect were collected between January and June 2022.

The study was approved both by the Ethics Committee of the University of Thessaly (ref. 130/08-02-2022) and the Scientific Council of the University Hospital of Larissa. It conformed to the Helsinki Guidelines for Research with Human Participants [25]. All participants signed a consent form.

### 2.2. Translation and Cross-Cultural Adaptation Process

The CRBS assesses CVD patient perceptions of the degree to which barriers at the levels of patients, healthcare providers and healthcare systems impose a negative effect on their enrollment and participation in CR programs. The original English version of the scale consists of 21 items. The CRBS has four subscales: healthcare system issues/perceived need, logistical factors, work/time conflicts, comorbidities or functional status [22]. Items are rated on a 5-point Likert-type scale, that ranges from 1 = strongly disagree to 5 = strongly agree; Higher scores indicate greater barriers to CR. The CRBS has been translated into 16 languages to date (see https://sgrace.info.yorku.ca/cr-barriers-scale/crbs-instructions-and-languages-translations/, accessed on 18 October 2022).

The original version of the scale was independently forward-translated from English into Greek by two translators, whose native language was Greek. One of the translators had a medical background. The two translations were grammatically assessed, and any discrepancies in the meanings and structures of the sentences and phrases were reconciled by consensus. This Greek synthesis version was backward-translated into English by a bilingual Greek-English native speaker, blinded to the original English version, resulting in the second version. The backward-translated version was linguistically and culturally reviewed, by an expert committee, comprising two CR experts and the engaged translators. Following the comparison of the revised version of the scale to the original version, and the mutual resolution of the conceptual discrepancies, the pre-final Greek version of CRBS (CRBS-GR) was harmonized Appendix A.

This pre-final version was field-tested in 15 cardiac outpatients from the University Hospital of Larissa. All 15 cardiac outpatients were native Greek speakers. The aim was to determine whether the translation was simple, appropriate and understandable using the test-retest method [26]. The investigators used semi-structured qualitative interviews to learn about patient perceptions of the scale, including any possible difficulties in understanding both the whole scale and its individual questions. Patients were asked to evaluate the different questions of the CRBS-GR version and their utility using a Visual Analogue Scale (VAS), where ‘0′ means not usable at all and ‘100′ means very usable. After cognitive debriefing and modification of any ambiguous expressions that were noted via discussion, the scale was ready for psychometric validation.

### 2.3. Psychometric Testing

In addition to administering the CRBS-GR in person, sociodemographic and clinical characteristics including sex, age, educational level, distance from residence to the hospital, body mass index (BMI), hyperlipidemia and tobacco use were collected via self-report. Some of these variables, as well as the depressive symptom scale, were used to assess their relationship with the subscales and concurrent validity.

For the assessment of concurrent validity, the Hospital Anxiety and Depression Scale (HADS) was used. The HADS is a 14-item self-report scale used to assess levels of anxiety and depression in general hospital outpatients [27]. The HADS consists of a 7-item anxiety subscale and a 7-item depression subscale. Item responses range from 0 (not at all) to 3 (most of the time). The sum of the seven items in each subscale is computed, with higher scores denoting greater symptoms. The reliability and validity of the Greek version of the HADS scale has been previously established [28].

Recruitment of eligible participants was conducted at the University Hospital of Larissa, using convenience sampling. Inclusion criteria were: adults after post-percutaneous coronary intervention (PCI), who were eligible for phase II CR [29], native Greek speakers, and with no cognitive deficits. Patients with implanted cardiac defibrillators or pacemakers, at significant acute cardiovascular risk or presenting orthopedic, neurological or mental disorders that prohibited the ability to exercise were excluded. One hundred and ten outpatients were sought, as the proposed sample size for performing factor analysis is at least five respondents for each item in the instrument being used [30].

The statistical package STATA 13.1 was used for data analysis. The level of significance for all tests was set at 0.05. Psychometric analyses were performed to assess the validity (construct and concurrent), internal consistency, and test-retest reliability of the CRBS-GR version [31]. A construct validity test was performed using factor analysis. The factorability of the 21 items was first assessed to determine the suitability of the data for factor analysis. This assessment included examining the correlation matrix, the Kaiser–Meyer–Olkin measure of sampling adequacy and Bartlett’s test of sphericity. After confirming suitability, factor extraction was performed using principal factors analysis. The number of components to retain was determined using the following criteria: (a) the rule of eigenvalues greater than 1 [32], (b) factors explaining at least 50% of the cumulative percentage of variance, (c) Cattell’s scree test [33], (d) factors with at least three variables with high factor loadings, and (e) the meaning of the variable loading on to the same factor. The obtained factors were rotated using the oblique Promax rotation, since the latter allows some correlation between the variables, with the factor loading scores set as significant at the value of 0.4 or higher. Finally, the retained components were assessed to ensure that they had at least three items with loadings greater than 0.4.

Regarding the construct validity of the scale, convergent and divergent validity besides revealing the correlation accordance of the variables with their own dimension, worked as well as another criterion for ensuring that the accurate number of factors have been retained. Concurrent validity was assessed by examining the correlation between the results of the HADS scale and the factors obtained from the factorial analysis [34].

The internal consistency was estimated by calculating Cronbach’s alpha coefficient for the scale and the subscales. Alpha values reflect the correlation of the items both within themselves and with the total score; values greater than 0.60 were considered acceptable [35]. Reliability was assessed via the intraclass correlation coefficient (ICC) using a test-retest method with an interval of three weeks [36] between the first and second completions of the CRBS-GR scale by 15 participants.

The relation of the scale to the patients’ characteristics (sex, age, lipid profile, distance, BMI, smoking and educational level) was evaluated through One-Way Analysis of Variance (ANOVA) and Chi-Squared tests as applicable, after confirming the normal distribution of the data (*p* > 0.05) using the Kolmogorov–Smirnov test.

## 3. Results

### 3.1. Translation and Cross-Cultural Adaptation

Through the process, it was determined that all 21 barrier items of the original English CRBS version were applied to the Greek context, and no additional barriers were warranted. Following translation and cross-cultural adaptation of the CRBS-GR, no need for major changes in the wording emerged.

The content and face validity of the CRBS-GR were established through the qualitative interviews with the 15 participants and their evaluation of the 21 questions/items of the CRBS-GR using the VAS scale. The participants completed the scale in a maximum time of 10 min. The results revealed a good level of clarity, simplicity and understanding of the CRBS-GR version for almost the entire sample (Table 1).

### 3.2. Psychometric Validation

A total of 110 participants completed the survey; Table 2 displays their characteristics.

Construct validity was investigated through principal factors analysis, as well as convergent and divergent validity. The Kaiser-Meyer-Olkin (KMO) value was 0.658 and Bartlett’s test was significant (*p* < 0.001), indicating the suitability of the data for factor analysis. Factor loadings were set at 0.4 to suppress all loadings less than the latter. Five factors with eigenvalues > 1 were extracted. However, since the last factor’s eigenvalue was almost equivalent to 1 (λ = 1.01651), possibly due to an overestimation of the number of factors [37], and based on the other aforementioned criteria including the Scree plot (Figure 1), the fifth factor was excluded. Consequently, four factors were extracted, accounting for 87.44% of the total variance. The first factor reflects “comorbidities/functional status”. The second factor reflects “logistical factors” such as transportation, distance, and cost. The third factor reflects “work constraints/lack of time”. The fourth factor reflects “perceived need/healthcare factors”. The eigenvalues and the percentage of variance explained by each factor are presented in Table 3. Convergent validity was supported in that factor loadings for 13 out of the 21 barriers (61.9%) were more than 0.4, and thus correlates more in their dimension. Divergent validity was supported in that 18 out of the 21 barriers (85.7%) correlated poorly to the factor scores computed for the other dimensions.

Cronbach’s alpha was used to examine the internal consistency, revealing a satisfactory overall reliability value (Cronbach Alpha = 0.70). Acceptable internal consistency was shown for all but one factor (Cronbach’s Alpha range = 0.56–0.74; Table 3).

The 3-week test–retest reliability of the CRBS-GR was also acceptable (ICC = 0.96, 95% confidence interval [CI]: 0.90–0.99). The ICC of the CRBS for the first two factors was 0.97 (95% CI: 0.92–0.99), for the third factor 0.91 (95% CI: 0.77–0.97) and the fourth factor 0.86 (95% CI: 0.64–0.95). The aforementioned results are shown in Appendix A.

One-Way ANOVA and Chi-Squared (*X*^2^) tests were conducted to assess the relationship and the differences between the patient characteristics and the CR barriers. Significant differences were revealed between sex and driving difficulty back to the hospital (*p* < 0.05), location of permanent residence and distance (*p* < 0.001), travel costs to/and from the hospital (*p* < 0.001) and driving difficulty back to the hospital (*p* < 0.05). Additionally, educational level was significantly correlated to the cost, the belief of exercise as tiring and prolonged waiting for CR referral and participation (*p* < 0.05), whilst hyperlipidemia was related to the belief of the unnecessariness of CR interventions (*p* < 0.05). Last but not least, age was significantly correlated with the distance, the beliefs of not needing CR and that the exercise is tiring, the frequent trips, the work commitments, the older age and the long time to catch up and enter a program (*p* < 0.05), as presented in Appendix A.

The concurrent validity of the CBRS-GR was assessed by evaluating the degree of correlation between the CRBS-GR and HADS [34]. Results showed moderate correlations of patient anxiety and depression levels (based on HADS) with the factors of “perceived need/healthcare”, “comorbidities and functional status”, “logistical” and “work constraints/lack of time” (Appendix A). The aforementioned results imply a small to moderate correlation between the participant psychological status and CR barriers; thus, confirming the concurrent validity of the CRBS-GR.

### 3.3. Main Barriers

The most frequent CR barriers detected were the distance from the hospital/rehabilitation center, travel costs, the lack of information about CR programs, and patient preference for home or community exercise, as opposed to the center. These barriers were found via the assessment of the median (=4) of the patient responses to the CRBS-GR.

## 4. Discussion

The aim of this study was to translate, cross-culturally adapt, and psychometrically validate the original English CRBS version into Greek. All 21 items of the original CRBS were considered appropriate to the Greek context, and no additional barriers were added. Face, content, construct, and concurrent validity were supported, as well as internal and test-retest reliability.

Factor analysis revealed four factors: comorbidities/functional status, logistical factors, work constraints/lack of time and perceived need/healthcare system issues. This is consistent with the original English version, the Malay, Persian and Czech versions [22,38,39,40]. Contrarily, the Brazilian, the Turkish and the Mandarin versions revealed five subscales [41,42,43] and the Korean version six subscales [44]. A fifth factor was also identified in the Greek version, presenting though an eigenvalue slightly greater than 1. Aiming to stay as close as possible to the original English version and taking into account several indicators, it was acceptably decided to eliminate this fifth factor. The difference lies with the perceived needs and the healthcare items being grouped into the same factor, whereas they are separate factors both in the Chinese/Mandarin version [42] and the Malay version [38], showing some cross-cultural variability based on context. Convergent and divergent validity scores were 61.9%% and 85.7%, respectively, revealing the good correlation of the variables in their corresponding factor.

Internal consistency for the whole scale and three of the four subscales was satisfactory. However, this perceived need/healthcare factor subscale had a Cronbach’s alpha value lower than 0.70.

Nevertheless, the test-retest reliability of the scale was also established with an ICC of 0.96 and was also high for all four subscales. The concurrent validity method was chosen to investigate and demonstrate the validity of the CRBS-GR version as related to the HADS questionnaire. Results showed a minor correlation between the aforementioned tools, thus implying the validity of the CRBS-GR version.

The most significant barriers reported by patients were “distance from the rehabilitation center”, “the cost of travel”, “the lack of information about CR programs” and “I already exercise at home or in my community”. The distance and the cost barriers also appear at first-ranking positions in other CRBS studies, such as those with the Turkish, Chinese and Persian translations [39,41,42]. Additionally, the barrier regarding patient preference for exercising at home rather than in a community center is in accordance with the results of the Canadian version [22]; possibly implying the underestimation of CR benefits and its comprehensive nature.

It needs to be mentioned that the home-based CR has been proven to be as safe [45,46,47], efficient and cost-effective as the center-based CR [48]. Moreover, the cardiac telerehabilitation, by intergrading information and communication technology (ICT) and wearable sensors technology in its implementation, appears as an alternative effective and safe CR intervention that could counter several CR barriers such as travel distance and costs [13,49,50].

## 5. Limitations

Several limitations warrant caution in interpreting the results of this study. The small sample size, the small percentage of female participants, inclusion of only stent patients and the fact that the sample recruitment took place from a single, university hospital, limit the generalizability of the study results. In addition, some patient characteristics that may impact CR use–such as marital status and disease severity—were not considered. Finally, the dearth of CR programs in Greece hindered the investigation of the criterion validity of the CRBS-GR version. More studies are needed among Greek-speaking cardiac patients to further validate and generalize the findings of this study.

## 6. Conclusions

The 21-item Greek version of the CRBS (CRBS-GR) has been demonstrated to have good psychometric properties, thus being a reliable and valid tool for the assessment and hopeful mitigation of CR barriers among the Greek-speaking cardiac population. It is hoped that having this scale for use, alongside efforts to develop more CR programs, can result in greater CR use, and hence better patient health outcomes.

## Figures and Tables

**Figure 1 ijerph-20-04064-f001:**
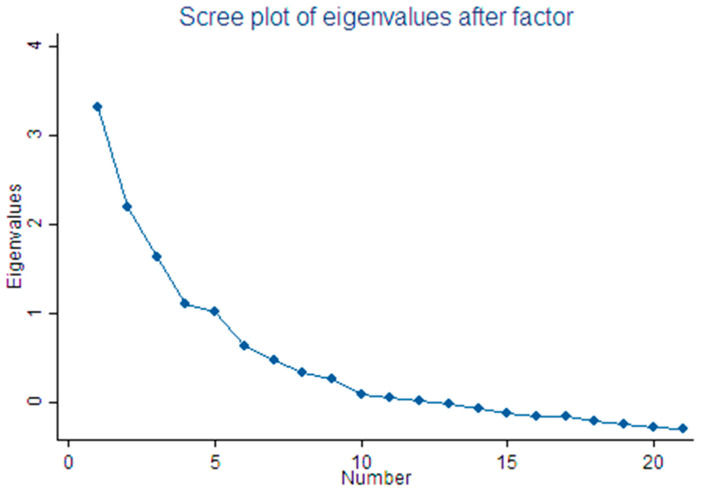
Scree plot of the eigenvalues and the number of factors.

**Table 1 ijerph-20-04064-t001:** Results from the subjective evaluation * of the CRBS-GR, N = 110.

1. Is the questionnaire, in your opinion, useful for assessing “barriers to cardiac rehabilitation”? (90.66%) 2. Do you think the questionnaire asks about your barriers to cardiac rehabilitation? (89.33%) 3. What do you think about the size of the questionnaire? (92.00%) 4. Are the questions clearly put forward? (87.33%) 5. Is the questionnaire well organized? (91.66%) 6. Did you understand the questions of the questionnaire? (92.33%) 7. Did you face any difficulty while completing the questionnaire? No (89.00%) 8. What do you think about the layout of the questionnaire questions? (88.00%).

CRBS-GR, cardiac rehabilitation barriers scale, Greek version; * visual analogue scale ranged from 0–100.

**Table 2 ijerph-20-04064-t002:** Characteristics of participants included in the study, N = 110.

Characteristic		
Age (years)	Mean = 65.27, SD = 10.21	
Gender	Male	97 (88.2%)
	Female	13 (11.8%)
Education level	Primary	56 (50.9%)
	Secondary	34 (30.9%)
	Higher	20 (18.2%)
Distance to center from home	Larissa or ≤50 km	52 (47.3%)
	>50 km	58 (52.7%)
Body mass index	Normal Weight	36 (32.7%)
	Overweight	46 (41.8%)
	Obese	27 (24.6%)
	Morbid Obesity	1 (0.9%)
Hyperlipidemia	Normal	27 (24.6%)
	Elevated	83 (75.5%)
Smoking	Yes	64 (58.2%)
	No	46 (41.8%)

SD, standard deviation; n (%) shown unless otherwise stated.

**Table 3 ijerph-20-04064-t003:** Factor Loadings * from Exploratory Factor Analysis (N = 110).

Items	Subscales			
	Comorbidities/Functional Status	Logistical Factors	Work Constraints/Lack of Time	Perceived Need/ Healthcare Factors
9. I find exercise tiring or painful	0.7309			
13. Lack of energy	0.6944			
15. Age (old)	0.5099			
14. Other health issues that prevent me from participating (specify: )	0.4641			
7. I already practice at home or in my community	−0.4061			
21. I prefer to take care of my health on my own and not through participating in groups	0.4002			
1. Distance (e.g., it is not located in your area, too far to travel)		0.8095		
2. Costs (e.g., parking, fuel)		0.7635		
3. Mobility difficulties (e.g., access in a car, Public transport)		0.5931		
8. Bad weather conditions		0.4233		
4. Family obligations		0.3366		
19. I think I got referred but the program did not contact me			0.7686	
20. It took me a long time to catch up and enter the program			0.7502	
11. Time constraints (e.g., very busy)			0.5062	
10. Frequent trips (e.g., holidays, business, home)			0.4674	
17. Many people with cardiac problems do not go and they are fine			0.4047	
12. Work commitments			0.3774	
16. My doctor didn’t think it was so necessary				0.5572
6. I don’t need cardiac rehabilitation				0.5133
5. Lack of information about cardiac rehabilitation (e.g., I did not have an update from my doctor)				0.5116
18. I can handle my heart problem on my own				0.3322
Eigenvalues	3.308	2.190	1.630	1.092
Variance explained (%)	35.19	23.30	17.34	11.61
Cumulative variance explained (%)	35.19	58.49	75.83	87.44
Reliability: Cronbach’s α	0.70	0.74	0.68	0.56

* Loadings above 0.40 are shown.

## Data Availability

Not applicable.

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
