# Peer review of "Translation, Cross-Cultural Adaptation and Psychometric Validation of the Greek Version of the Cardiac Rehabilitation Barriers Scale (CRBS-GR): What Are the Barriers in South-East Europe?"

_ijerph, 2023, doi:10.3390/ijerph20054064_

Round 1
Reviewer 1 Report
I would like to thank for the opportunity to review the manuscript of Antoniou et al. "Translation, Cross-Cultural Adaptation, and Psychometric Validation of the Greek version of the Cardiac Rehabilitation Barriers Scale (CRBS-GR): What are the barriers in South-East Europe?". In this article, the authors developed the Greek version of the Cardiac Rehabilitation Barriers Scale, carried out its cross-cultural adaptation, psychometric testing and validation. Without a doubt, this scale can contribute to a more frequent use of cardiorehabilitation in Greece, so the practical benefit of this article is beyond doubt.
However, when reviewing the manuscript, I had questions and comments to which I would like to receive answers from the authors.
1. I would like to clarify the characteristics of the patients included in the study. Why did the authors not take into account the presence of a family (married or not?), the severity of clinical symptoms (angina pectoris of heart failure)? In my opinion, these indicators may have a greater impact on participation in rehabilitation programs than the presence / absence of dyslipidemia.
2. It is not clear what was the participation of patients in the second stage of rehabilitation, what components did it include? In the Restrictions section, the authors stated that "the lack of CR programs in Greece obstructed the investigation of the criterion validity of the CRBS-GR version". So after all, are there any cardiorehabilitation programs in Greece or not? And the second question - if they are not, then how can patients assess the barriers to participation in them?
3. I would like to understand why exactly 3-week test-retest interval was chosen to evaluate internal consistency, not a longer interval (for example, 3 months [1]).
4. In the Limitations section, the authors indicated that "the participants may have responded in a socially-desirable manner to the HADS scale, as it was administered on the phone". Does this mean that the Cardiac Rehabilitation Barriers Scale data was also obtained over the phone? Then it should have been specified in the Methods section. If this data was obtained in person, then why couldn't the HADS questionnaire be administered in person as well?
5. I think that among the limitations it should be noted that the study included patients only after the PCI procedure. In fact, cardiac rehabilitation programs involve a large number of patients with other conditions (after myocardial infarction, after open heart surgery), which may have other options for barriers to participation in these programs.
References:
1. Pushkarev, G. S., Zimet, G. D., Kuznetsov, V. A., & Yaroslavskaya, E. I. (2020). The Multidimensional Scale of Perceived Social Support (MSPSS): Reliability and Validity of Russian Version. Clinical gerontologist, 43(3), 331–339. https://doi.org/10.1080/07317115.2018.1558325
Author Response
Reviewer #1:
- I would like to clarify the characteristics of the patients included in the study. Why did the authors not take into account the presence of a family (married or not?), the severity of clinical symptoms (angina pectoris of heart failure)? In my opinion, these indicators may have a greater impact on participation in rehabilitation programs than the presence / absence of dyslipidemia.
RESPONSE: It is unfortunate we did not capture these characteristics; we wanted to minimize respondent burden. Your important point has been added to the study limitations.
It is not clear what was the participation of patients in the second stage of rehabilitation, what components did it include? In the Restrictions section, the authors stated that "the lack of CR programs in Greece obstructed the investigation of the criterion validity of the CRBS-GR version". So after all, are there any cardiorehabilitation programs in Greece or not? And the second question - if they are not, then how can patients assess the barriers to participation in them?
RESPONSE: As stated in the introduction, there were 4 programs in Greece in 2016; it is likely this is higher now. In the limitations, we revised “lack of” to “dearth” to be accurate. There is a CR program in Larissa where the study took place (see: https://bmjopen.bmj.com/content/12/6/e059945.abstract).
The participant inclusion criteria has been clarified to state that participants were to be eligible for CR. We have added a citation to describe the program.
The CRBS assesses barriers to enrolment as well as to adherence to sessions. Some of the items relate to lack of proximate programs as you see. This is why in the conclusions we state that structural changes must be made to increase program availability along with mitigating the other barriers identified.
I would like to understand why exactly 3-week test retest interval was chosen to evaluate internal consistency, not a longer interval (for example, 3 months [1]).
RESPONSE: Barriers could change over that long a period of time, as we have shown in other studies responsiveness of the CRBS to exposure to different contexts or interventions. Studies show that the optimal time interval between testing varies depending on the construct being measured on the stability of the construct over time and on the target population. In fact, an even shorter 2 weeks is often used. We have added a methods citation to support the 3 week interval
In the Limitations section, the authors indicated that "the participants may have responded in a socially desirable manner to the HADS scale, as it was administered on the phone". Does this mean that the Cardiac Rehabilitation Barriers Scale data was also obtained over the phone? Then it should have been specified in Methods section. If this data was obtained in persion, then why couldn’t the HADS questionnaire be administered in person as well?
RESPONSE: It is now clarified that the surveys were administered in person.
- I think that among the limitations it should be noted that the study included patients only after the PCI procedure. In fact, cardiac rehabilitation programs involve a large number of patients with other conditions (after myocardial infarction, after open heart surgery), which may have other options for barriers to participation in these programs.
RESPONSE: This has been added to the limitations.
Reviewer 2 Report
Dear Authors,
Thank you for your work is this very important topic. Your study is well designed, and your paper is well-written. My concern is regarding your sample. You mentioned in your introduction that women are under-represented in cardiac rehab. They are also under-represented in your study. Why is the rate of female participants so low? Also, in table 2 how does your sample compare to the overall cardiac population, and the cardiac rehab population in Greece?
Author Response
Thank you for your work is this very important topic. Your study is well designed, and your paper is well written. My concern is regarding your sample. You mentioned in your introduction that women are under-represented in cardiac rehab. They are also under-represented in your study. Why is the rate of female participants so low? Also, in table 2 how does your sample compare to the overall cardiac population, and the cardiac rehab population in Greece?
RESPONSE: Indeed, this is a pervasive issue worldwide, that has been well-studied, but is beyond the scope of this paper. If you are interested, here are some nice reviews: https://www.ncbi.nlm.nih.gov/pmc/articles/PMC8243714/; https://pubmed.ncbi.nlm.nih.gov/28388314/.
The mean age of CHD patients in Greece is slightly older than our sample (65) at 70 years. There are only ~4 CR programs in Greece, so as you can imagine there is not much data on the characteristics of the cardiac rehabilitation population. 47% of Greek adults smoke; thus the 58% rate in our sample of PCI patients is reasonable. Similarly 17% of Greek adults are obese, so the 25% rate in our PCI sample is again not unreasonable. Nevertheless, there are caveats about generalizability in the limitations section, with a call for more needed research.
Round 2
Reviewer 1 Report
The authors corrected the manuscript, they answered my comments and questions. I have no other comments.